# Can the *FUT 2* Gene Variant Have an Effect on the Body Weight of Patients Undergoing Bariatric Surgery?—Preliminary, Exploratory Study

**DOI:** 10.3390/nu12092621

**Published:** 2020-08-28

**Authors:** Natalia Komorniak, Alexandra Martynova-Van Kley, Armen Nalian, Wiktoria Wardziukiewicz, Karolina Skonieczna-Żydecka, Daniel Styburski, Joanna Palma, Bartosz Kowalewski, Krzysztof Kaseja, Ewa Stachowska

**Affiliations:** 1Department of Human Nutrition and Metabolomics, Pomeranian Medical University in Szczecin, 71-460 Szczecin, Poland; natalia.komorniak@wp.pl (N.K.); wiktoria.wardziukiewicz@gmail.com (W.W.); karzyd@pum.edu.pl (K.S.-Ż.); zbizcz@pum.edu.pl (D.S.); palma.01.01@gmail.com (J.P.); 2Department of Biology, Stephen F. Austin State University, Nacogdoches, TX 75962, USA; avankley@sfasu.edu (A.M.-V.K.); biology@sfasu.edu (A.N.); 3Independent Provincial Public Hospital Complex in Szczecin—Zdunowo, 70-891 Szczecin, Poland; spwsz@spwsz.szczecin.pl; 4Department of General Surgery and Transplantation, Independent Public Clinical Hospital No.2, 70-111 Szczecin, Poland; kaseja@autograf.pl

**Keywords:** bariatric surgery, obesity, Roux-en-Y gastric bypass, sleeve gastrectomy, microbiota, short-chain fatty acids, *FUT2* gene, propionate, waist-hip ratio

## Abstract

Background: The *FUT2* gene (Se gene) encoding the enzyme α-1,2-L-fucosyltransferase 2 seems to have a significant effect on the number and type of bacteria colonizing the intestines. Methods: In a group of 19 patients after bariatric surgery, the polymorphism (rs601338) of *FUT2* gene was analyzed in combination with body mass reduction, intestinal microbiome (16S RNA sequencing), and short chain fatty acids (SCFA) measurements in stools. Results: Among the secretors (Se/Se polymorphism of the *FUT2* gene rs601338, carriers of GG variant), correlations between waist-hip ratio (WHR) and propionate content and an increase in *Prevotella*, *Escherichia*, *Shigella*, and *Bacteroides* were observed. On the other hand—in non-secretors (carriers of GA and AA variants)—higher abundance of *Enterobacteriaceae*, *Ruminococcaceae*, *Enterobacteriaceae*, *Clostridiales* was recorded. Conclusions: The increased concentrations of propionate observed among the GG variants of *FUT 2* may be used as an additional source of energy for the patient and may have a higher risk of increasing the WHR than carriers of the other variants (GA and AA).

## 1. Introduction

The number of patients with severe obesity is gradually increasing. With relation to conservative treatments, bariatric surgery is considered to be a more effective way of treating patients with severe obesity, leading to reduction in both weight and coexisting diseases and mortality [1]. According to the American Society for Metabolic and Bariatric Surgery, the number of metabolic and bariatric procedures performed in 2018 was 10.8% higher than in 2017; only approximately 1% of all patients who qualify for bariatric surgery actually undergo surgery [2].

One of the factors that may be involved in the pathogenesis of severe obesity is intestinal microbiota. The mechanism includes changes in the composition of intestinal bacteria described as intestinal dysbiosis (disturbed relation between *Bacteroidetes* and *Firmicutes* types) generating inflammation [3].

Systematic reviews have shown that bariatric surgeries affect the human microbiome. Microbial richness increases significantly within 3 months and is maintained after 1 to 2 years post-surgery. Greater alterations in the microbiome occur after Roux-en-Y gastric bypass, than after purely restrictive bariatric procedures (e.g., sleeve gastrectomy). An increase in Bacteroidetes and Proteobacteria and a decrease in Firmicutes post-bariatric surgery are observed. These changes induce an increase of the Bacteroides/Firmicutes ratio, which are positively associated with weight loss post-bariatric surgery [4].

The *FUT2* gene, or Se gene, codes the enzyme α-1,2-L-fucosyltransferase 2 and conditions the formation of H antigens in the intestinal epithelial tissue [5]. Individuals with at least one Se allele (Se/Se or Se/se genotypes) have ABO and H system antigens on epithelial cells and are called secretors (variant G428A (se428, W143X, rs601338). In contrast, homozygotes for *FUT2* AA (se/se, The *FUT2-*AAA (G428A, W143X)), are called non-secretors [6,7,8], which means they are unable to secrete histo-blood group antigens into bodily fluids, or express them on mucosal surfaces [9]. *FUT2* (rs601338, G428A) is the most common polymorphism in *FUT2* Caucasian non-secretors [5]. When the *FUT 2* gene is expressed in the intestines, the antigens encoded by this gene participate in the anchorage of various intestinal bacteria [10]. It seems that the type of the *FUT2* gene allele may be one of the main determinants of bacteria colonizing the intestines [8].

Studies have shown that *FUT2* gene expression may be influenced by intestinal microbiota, e.g., in response to infection risk [5,11,12]. Symbiotic bacteria may induce a signaling cascade leading to increased expression of the *FUT2* gene, thus modulating colonization of the gastrointestinal tract by other microorganisms [12].

Comparing the overall number and diversity of intestinal microbiota, it is indicated that non-secretors have significantly lower diversity (the number of microbial genes was less by 15% on average) [13]. The basic difference between Se/Se and se/se genotypes is the number of *Bifidobacterium* bacteria. This genus is significantly less abundant among non-secretors (AA). Among the non-secretors, a lower number microorganisms producing short chain fatty acids, including anti-inflammatory butyrate (e.g., *Roseburia intestinales*), was also identified [13,14]. In addition, a different number of *Lactobacillus* and *Clostridium* bacteria is observed in secretors compared to non-secretors [15].

Carbohydrates that are not digested in the stomach are transferred to the small and large intestine, where they are used by intestinal microbiota that ferment them to short chain fatty acids (SCFA). Acetate, propionate, and butyrate are produced in the intestines. SCFA can act as an intermediary in controlling energy consumption. Activation of the orphan GPR41 receptor by propionate can have an appetite suppressant effect as propionate stimulates leptin synthesis [16,17].

The aim of the study was to determine whether variants of the *FUT2* gene may be responsible for differences in the profile of intestinal microbiota in patients undergoing bariatric surgery who developed depressive disorders. The hypothesis we wanted to verify included the assumption that people with variants of Se/Se and Se/se of the *FUT2* gene through increased number of bacteria involved in the SCFA synthesis process and probable higher SCFA synthesis will have higher body weight compared to “non-secreting” people (se/se *FUT2*) [18].

## 2. Materials and Methods

### 2.1. Study Area

The study was attended by patients who were after bariatric surgery and had depressive disorders. In the literature, there are reports of the development of mental disorders in some patients after surgery [19,20], although this group has been relatively little studied.

Patients were recruited at the Surgical Obesity Surgery Outpatient Clinic in Szczecin Zdunowo, Poland between July 2018 and March 2019. At the outpatient clinic, all patients (*n* = 117) who underwent Sleeve gastrectomy or Roux-en-Y gastric bypass surgery in at least the last 6 months were asked to fill in the Beck scale. In this study, out of 117 interviewed individuals, 49 met the criteria for inclusion (Beck scale ≥12 points), of which 19 were women (Figure 1). This was an exploratory study without a prior power calculation.

Individuals qualified for the study were enrolled in the Department of Human Nutrition and Metabolomics of Pomeranian Medical University in Szczecin and a fecal sample was collected and frozen at −80 °C until analysis. On-site anthropometric measurements, a food frequency questionnaire supplemented with questions on the type of bariatric surgery, weight reduction after surgery, the supplementation used, and drugs used, as well as the occurrence of diseases and ailments (i.e., abdominal pain, nausea, vomiting, heartburn, constipation, diarrhea) were given. Additionally, the patient’s dietary reports from 72 h were collected and analyzed with the use of the 5D diet program. In order to obtain material for genetic testing, a blood sample was taken.

All subjects gave their informed consent for inclusion before they participated in the study. The study was conducted in accordance with the Declaration of Helsinki, and the protocol was approved by the Ethics Committee of Pomeranian Medical University in Szczecin (No KB-0012/40/17). A prospective observational study was carried out in individuals who were at least 6 months after bariatric operation (performed with the sleeve gastrectomy (SG) or Roux-en-Y gastric bypass (RYGB)). The inclusion criterion for the study group included the determination of depressive disorders using the Beck scale (≥12 points) [21,22]. All participants declared that they did not take antidepressants. The exclusion criteria included the use of non-steroidal anti-inflammatory drugs, antibiotics, probiotics, and proton pump inhibitors during the 6 months preceding the start of the study.

### 2.2. Study Protocol, the Anthropometric Data

During the check-up visit, anthropometric data were collected. Using a medical tape, hip and waist circumference was measured. Based on these measurements, waist-hip ratio (WHR) was calculated (WHR = waist circumference (cm)/hip circumference (cm)), body mass index (BMI) was calculated according to these measurements (Body mass index BMI = body weight (kg)/square of height (m)) (Table 1 and Figure 2).

### 2.3. DNA Isolation and Real-Time PCR SNP Analysis

A method based on quantitative DNA polymerase chain reaction (qPCR) was used to evaluate the polymorphism rs 601338 of the *FUT2* gene. The genetic material was amplified in real-time polymerase chain reaction with LightCycler^®^96 (Roche). Genotyping was performed according to the TaqMan method using 10 ng of DNA. The reaction mixture was TaqMan Genotyping Master Mix (catalogue number: 437135, Thermo Fisher Scientific, Foster City, CA, USA) and oligonucleotide TaqMan SNP Genotyping Assays (Applied Biosystems, Thermo Fisher Scientific, Foster City, CA, USA).

### 2.4. DNA Extraction and the Analysis of Bacterial 16S RNA Gene Sequencing

DNA extraction and sequencing of the regions V1–V2 of the 16S r DNA gene on Illumina MiSeq (Illumina Inc, San Diego, CA, USA) were performed at the Institute of Clinical Molecular Biology in Kiel University (Kiel, Germany) by their in-house protocol. DNA isolation was carried out using the QIAamp DNA Fast Stool Mini Kit automated on the QIAcube (Qiagen, Hilden, Germany). The samples were transferred to 0.70 mm Garnet Bead Tubes (Dianova, Hamburg, Germany) and were subjected to 1.1 mL ASL lysis buffer and bead beating using the a SpeedMill PLUS (Analytik Jena, Jena, Germany) for 45 s at 50 Hz. After this step, samples were heated to 95 °C for 5 min. Further analysis proceeded according to the manufacturer’s recommendations. The acquired DNA concentration was 10–40 µg/sample.

Analyses of bacterial 16S RNA were based on the amplification of variable regions V1 and V2 using the primer pair 27F-338R in a dual-barcoding approach according to Caporaso et al. [23]. For amplification, we applied 3 µL of DNA, dilution 1:10. PCR products were analyzed by agarose gel electrophoresis and standardized by SequalPrep Normalization Plate Kit (Thermo Fischer Scientific, Waltham, MA, USA). Sequencing was performed using the Illumina MiSeq v3 2 × 300 bp (Illumina Inc., San Diego, CA, USA), and demultiplexing was based on 0 mismatches in the barcode sequences. The analysis was performed using FLASh software, which allowed for the overlapping of the reads between 250 and 300 bp [24]. Chimeras were removed using UCHIME [25], and sequences with low quality were eliminated (less than 5% of total sequences).

### 2.5. Sequence Statistical Analysis

Due to the small sample size, nonparametric analyses were adopted as appropriate. To analyze whether the FUT genotype affects weight loss, a Kruskal–Wallis test was used.

Assembly of Illumina overlapping paired-end reads was done with PANDAseq [26]. The sequences were identified taxonomically using the RDP classifier 2.11 [27]. Community structure analyses were based on the phylum and genus taxonomic levels.

Principal component analysis (PCA), non-metric multidimensional scaling (NMDS), and redundancy analysis (RDA) ordinations on Bray–Curtis dissimilarities were performed in R- 3.5.1 [28], package (VEGAN). All permutation tests were performed with 10,000 permutations. Constrained ordination was performed to test for correlation between bacterial community structure and anthropometric factors.

## 3. Results

### 3.1. Study Group Characteristics

Among the patients, a significant weight reduction after the surgery was noted (*p* = 0.0001) (Figure 3). There was no significant correlation between the *FUT2* gene variant and BMI reduction (*p* = 0.77) (Table 2).

Genetic examination showed that the majority of patients were *FUT2* genes secretors (GA 47.4% and GG 42.1%). Only 10.5% of the patients were non-secretors. In the study group, gastrointestinal complaints persisted, which included: problems with emptying (26.3% of people suffered from abdominal pain and recurrent diarrhea, while 36.8% had constipation), 26.3% had nausea and vomiting, and 36.8% felt heartburn.

Analysis of 72 h diaries showed a very low dietary fiber intake in the study group (15.5 ± 6.4 g). At the same time, the meals consumed by the patients were rich in protein, 65 ± 16.2 g (with a predominance of animal protein over plant protein 44.89 ± 14 vs. 14.87 ± 5.68 g). The result of the analysis of dietary logs is presented in Table 3.

### 3.2. Characteristics of the Stool Microbial Composition

Taxonomic compositions of the microbiota obtained from all the subjects were analyzed at the phylum and genus levels, expressed as operational taxonomic units (OTUs) (Table 4). Three phyla were present in all patients Firmicutes (22.78%), Bacteroidetes (60.75%), and Proteobacteria (16.47%).

Additionally, there were clear differences in the composition of the microbiome depending on the *FUT2* gene variant (Figure 4, Figure 5, Figure 6 and Figure 7).

Interestingly, there was a significant correlation (*p* = 0.01) between abundance of OTU’s with two categorical factors: *FUT2* gene variant (GA, AA, GG) and intolerance and two continuous variables: waist-to-hip ratio (WHR) and propionic concentration. GG *FUT2* carriers had a high *Prevotella* and *Escherichia*. *Shigella* content increased with propionic concentration. In addition, in GG carriers, *Bacteroides* abundance increased with WHR. In turn, carriers of variant AA and GA had more *Enterobacteriaceae*, *Ruminococcaceae*, *Enterobacteriaceae*, *Clostridiales*, and *Bacteroidetes* than GG (Figure 8).

## 4. Discussion

It is estimated that 5–10% of daily calories come from SCFA oxidation, and their metabolites can be used for the synthesis of de novo lipids and glucose [29]. Both epidemiological studies and animal models have shown that higher SCFA concentrations in feces are positively correlated with body weight [30,31,32] and the use of high-energy diets [33]. Moreover, metagenome studies on obesity have shown that the intestinal microbiome of an obese man was enriched with pathways involved in microbiological processing of carbohydrates (e.g., phosphotransferase pathways), as well as genes involved in the production of SCFA (e.g., acetyl/propionyl-CoA carboxylase and acetyl-CoA carboxylase) [34]. Interestingly, after performing the Roux-en-Y gastric bypass bariatric surgery (related to weight reduction in patients), an increase in the colon propionate content is observed [35]. It should be remembered that the effect of weight reduction and metabolic changes after bariatric procedures is associated not only with a reduction in the amount of food consumed, but (perhaps above all) with a change in intestinal microbiota and modification of the secretion of intestinal hormones [36]. It has been shown that increased concentration of propionate stimulates the secretion of anorexygenic intestinal peptides—YY peptide hormone (PYY) and glucagon-like peptide-1 (GLP-1) from colon cells, which in turn affects the feeling of satiety [37] and reduces the risk of weight gain.

In our study, however, we obtained the opposite results. Among the carriers of the GG variant of the FUT2 gene, we observed an increase in both propionate and WHR levels (Figure 8). At this point, it is worth mentioning that the BMI (BMI of our patients was 30.6 ± 5.3 kg/m^2^) is often used (due to its simplicity and low cost) to assess body weight. However, the main disadvantage is the fact that it does not allow one to determine the body composition (the proportion between fat and fat-free body weight). In the context of health risks associated with excess body fat (especially metabolically active visceral fat), WHR is a more precise indicator. Studies have shown that WHR is a better predictive factor for visceral fat, cardiometabolic diseases, and mortality than BMI [38]. Therefore, the results of our research can be extremely important in the context of the effectiveness of body fat reduction after bariatric surgery.

In this study, the numbers of *Prevotella*, *Escherichia*, *Schigella*, and *Bacteroides* increased with the increase in propionate and WHR. This is an interesting observation, as it is the reduced ratio of Bacteroidetes to Firmicutes that is described as characteristic for obese individuals [39]. In addition, studies of patients after bariatric surgery report a decrease in the number of *Firmicutes* and an increase in *Proteobacteria*, *Bacteroidetes*, and *Bacteroidetes*/*Firmicutes* ratio. Additionally, in this group of patients, the increase in the amount of *Bacteroidetes* correlates with fat reduction and weight loss [4,40]. The increase in the amount of *Escherichia coli*, in turn, is associated with greater efficiency in obtaining energy in a state similar to starving after RYGB [41]. A frequently described phenomenon after bariatric surgeries is also a decrease in the amount of *Bifidobacterium* [4]. It seems that a microbiome rich in *Bifidobacterium* bacteria favors the preservation of homeostasis and the maintenance of normal body weight (which has been confirmed in studies on animal models and in clinical studies on the human population) [42,43,44]. The reason for the correlation with the increase in the WHR presented in our study may be the increased production of short-chain fatty acids and their use as an additional source of energy, as well as their indirect effect on metabolism. It has been shown that there are strains of *Bifidobacterium*, e.g., *B. longum*, which have the ability to convert lactose to butyric acid [45]. In addition, a higher abundance of *Roseburia intestinales* was found among the secretors, which is also responsible for the intensive synthesis of this acid [13,14]. The majority of butyrate (about 70–90%) is used by colonocytes as an energy source, however, the remaining amount is metabolized by the host and constitutes an additional energy input obtained from potentially indigestible sources [46]. Additionally, short-chain fatty acids can stimulate weight gain and adipogenesis in white adipose tissue by binding to FFAR2 and FFAR3 (free fatty acid receptor) [47]. Interestingly, in our study, a similar relationship was not found in carriers of the remaining variants of the *FUT2* gene, which suggests that *FUT2* secretors in the dominant homozygous type have a higher risk of ineffectiveness of bariatric surgery in fat reduction than carriers of the recessive allele.

It is also important that although the patients were on average 3.2 years after surgery, their BMI and WHR indices still indicated visceral obesity (I degree) and depressive disorders. The diet used by the patients—high-protein, at the same time poor in dietary fiber, changes the characteristics of bacterial fermentation and leads to intestinal dysbiosis [48]. Symptoms occurring in almost 30% of our patients (such as recurrent abdominal pain, diarrhea, constipation) may be a manifestation of disorders within the intestinal microflora that was previously described after RYGB [49].These disorders negatively impact health-related quality of life because of effects on gastrointestinal motility, visceral hypersensitivity, changes in gut permeability, immune activation, gut–brain dysregulation, and central nervous system dysfunction [50]. In dysbiosis, the integrity of the intestinal barrier (leaky gut syndrome), endotoxemia (i.e., penetration of fragments of the cell wall of Gram-negative bacteria—lipopolysaccharide first to the area of the intestinal mucosa proper plate and then to circulation) are disturbed. This phenomenon generates a state of local microinflammation (intestinal mucosa) which is dynamically transmitted throughout the body. It is the chronic low-intensity inflammation that is responsible for the formation and progression of obesity, insulin resistance, and generation of liver steatosis. Other mechanisms binding bacterial dysbiosis to obesity are negative influences on the speed of intestinal passage, which makes it easier for bacteria to obtain energy from intestinal contents more efficiently (in obese people, an increase in SCFAs content in stool, decrease in lipolysis, and intensification of lipogenesis were found) [30].

## 5. Conclusions

The study suggests that the *FUT2* gene variant may have a significant impact on the effectiveness of visceral fat loss after bariatric surgery. Secretors (GG) as a result of increased levels of propionate, which can be used as an additional source of energy for the body, may be less effective in reducing their WHR than carriers of the other variants. Special attention should be paid to the balanced diet of patients after surgery, so that apart from adequate protein, they also eat products rich in vitamins, minerals, and fiber, which may reduce the risk of developing intestinal dysbiosis. Undoubtedly, there is a need for further research on this subject.

## 6. Limitations

This study has several limitations worth noting. Although we stated that the number of patients who met the inclusion criteria were 49, only 19 people were enrolled in this research. This undoubtedly makes it necessary to repeat the study on a larger group of participants. Second, the patients were recruited at the Surgical Obesity Surgery Outpatient Clinic in Szczecin Zdunowo, but due to its outpatient nature, patients who have been operated on in different hospitals may be treated here (in this research, 11 patients were operated on by the two surgeons from Zdunowo Hospital and 8 patients by other surgeons). Thirdly, although all patients underwent laparoscopic SG or RYGB bariatric surgery, they were operated on by different surgeons, which creates a risk of potential differences in surgical techniques that could affect the results of this study.

This was an exploratory study without a prior power calculation. Therefore, it is necessary to confirm these findings in future properly designed studies.

## Figures and Tables

**Figure 1 nutrients-12-02621-f001:**
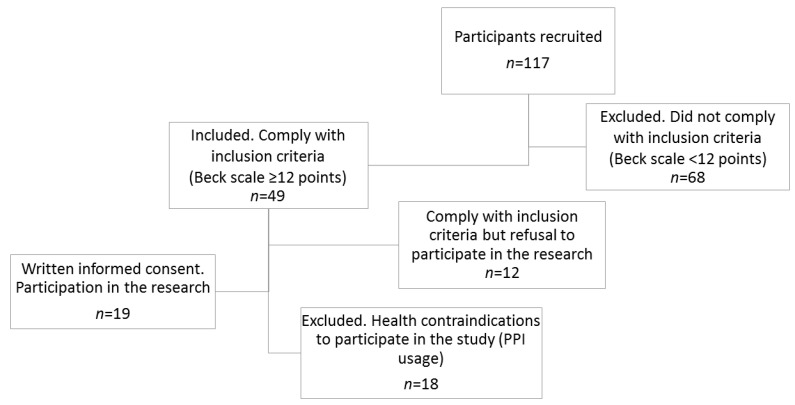
Scheme of study group selection.

**Figure 2 nutrients-12-02621-f002:**
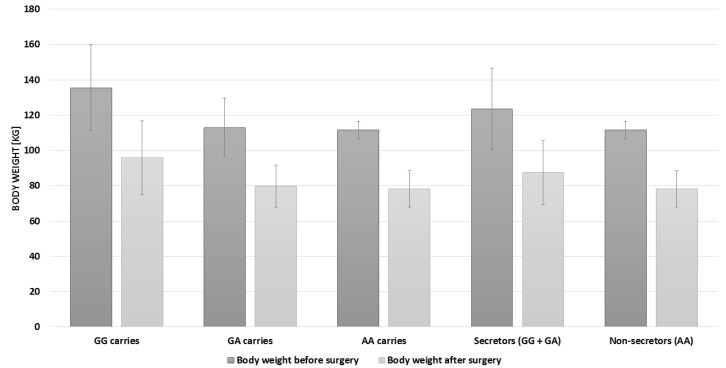
Body weight of *FUT2* variants.

**Figure 3 nutrients-12-02621-f003:**
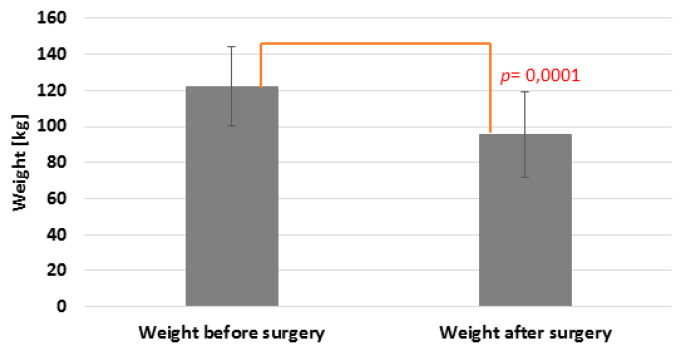
Weight reduction after bariatric surgery.

**Figure 4 nutrients-12-02621-f004:**
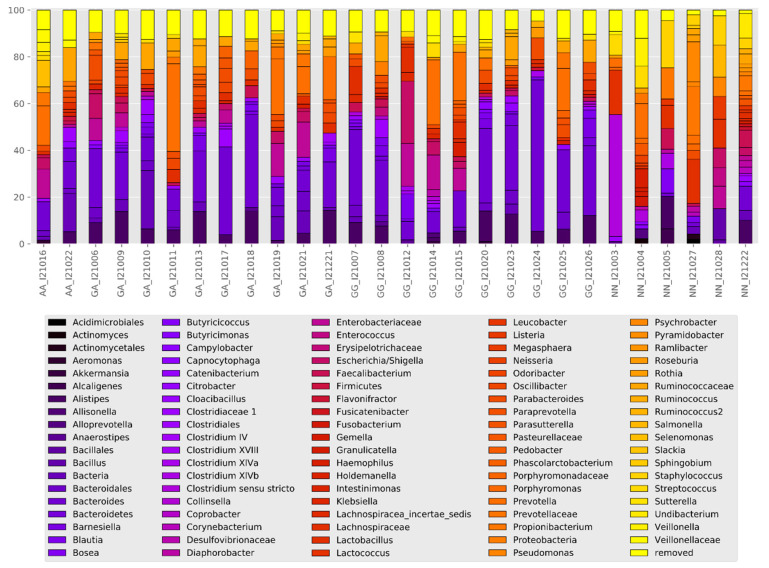
Stool microbiota composition.

**Figure 5 nutrients-12-02621-f005:**
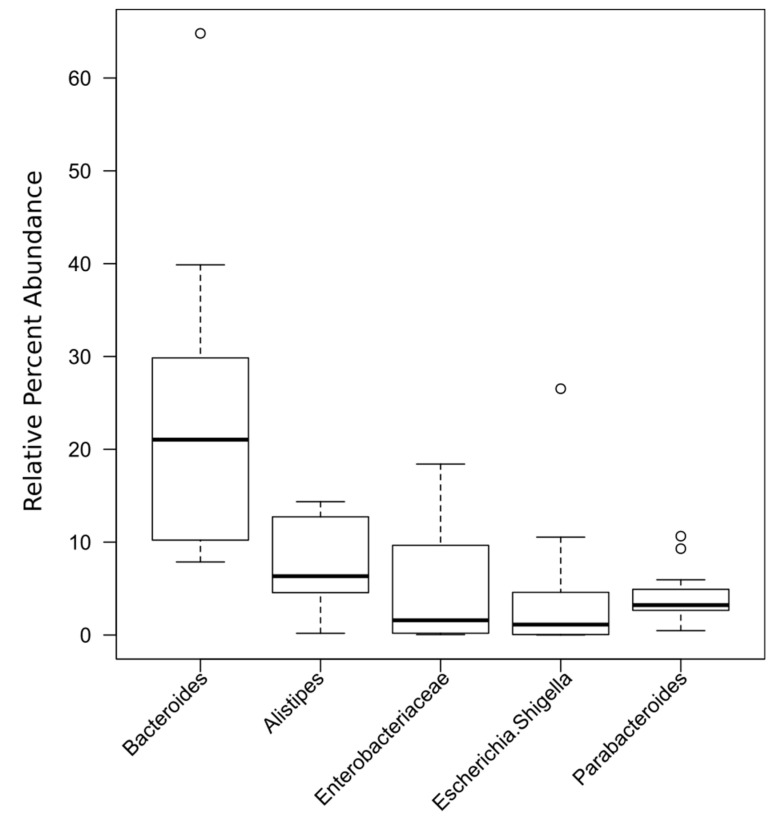
Boxplot of most abundant taxa from all the samples.

**Figure 6 nutrients-12-02621-f006:**
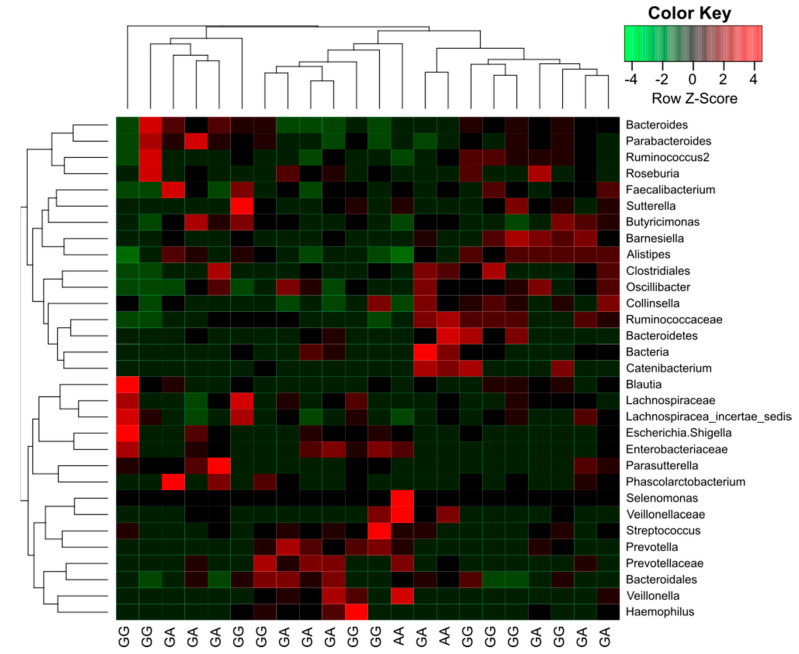
Associations between changes in microbial composition and gene *FUT2* variants.

**Figure 7 nutrients-12-02621-f007:**
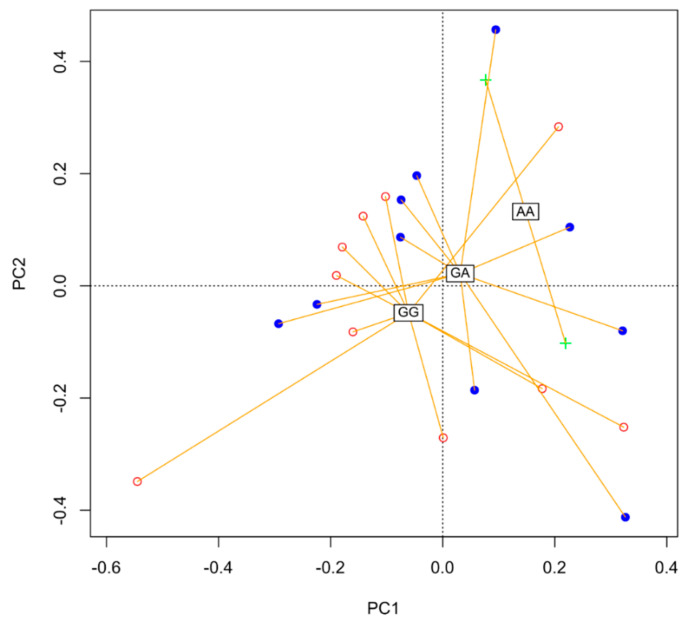
Differences in the composition of the microbiome depending on the *FUT2* gene (principal coordinates analysis (PCoA)).

**Figure 8 nutrients-12-02621-f008:**
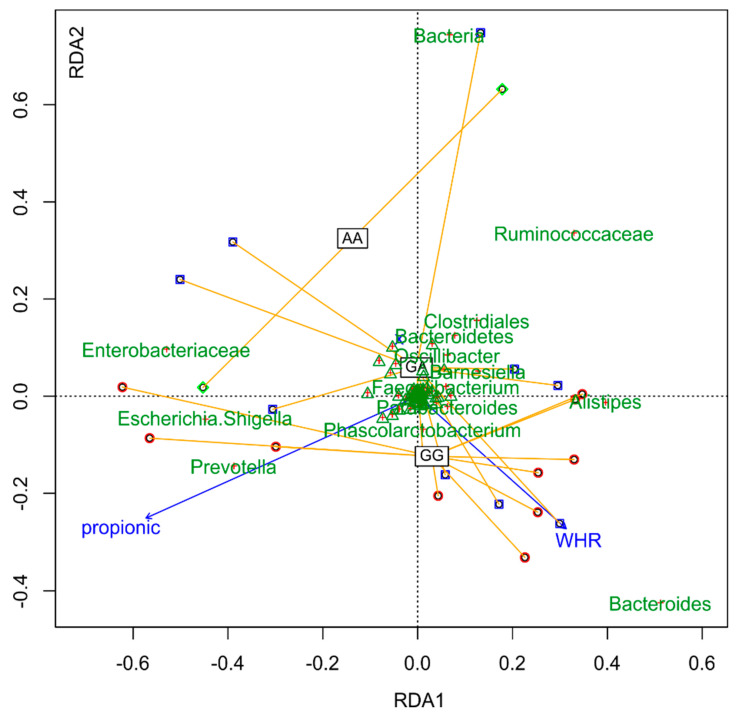
Factors determining differences between samples (distance-based redundancy analysis (db-RDA)).

**Table 1 nutrients-12-02621-t001:** Characteristics of the study group.

Feature	
Age (years)	46.3 (±11.7)
Type of surgery	10 patients Roux-en-Y gastric bypass (RYGB) 9 patients sleeve gastrectomy (SG)
Weight before surgery (kg)	122.3 (±22.1)
BMI before surgery (kg/m^2^)	43.2 (±5.9)
Post-operative period (years)	3.2 (±3.4)
Actual weight (kg)	95.5 (±23.5)
Actual BMI (kg/m^2^)	30.6 (±5.3)
Actual waist circumference (cm)	94.2(±12.2)
Actual WHR	0.82 (±0.07)

± standard deviation.

**Table 2 nutrients-12-02621-t002:** ΔBMI after surgery in case of *FUT2* variants.

Fut2	*n*	Delta-BMI	SD
AA	2	−13.4000	3.1113
GA	9	−12.1571	4.9784
GG	8	−13.8023	4.5286
Significance level			*p* = 0.77

SD: standard deviation. AA, GA, GG: *FUT2* gene variants

**Table 3 nutrients-12-02621-t003:** Seventy-two-hour nutritional diary log summary.

**Energy (kcal)**	**Total Protein (g)**	**Animal Protein (g)**	**Plant Protein (g)**	**Fat (g)**	**Carbohydrates (g)**
1138 ± 366	65 ± 16.2	44.89 ± 14	14.87 ± 5.68 g	45.2 ± 18.8	117.6 ± 44.9
**Fiber (g)**	**Sucrose (g)**	**Sodium (mg)**	**Potassium (mg)**	**Calcium (mg)**	**Magnesium (mg)**
15.5 ± 6.4	19.7 ± 12.4	1144 ± 544.5	2622 ± 774.4	497.2 ± 152.4	256.2 ± 88.5
**Iron (mg)**	**Phosphorus (mg)**	**Zinc (mg)**	**Copper (mg)**	**Vitamin A (µg)**	**Vitamin D (µg)**
8.9 ± 2.6	1008.3 ± 231.3	7.1 ± 1.8	1.03 ± 0.3	1311.9 ± 1671	3.2 ± 3.3
**Vitamin E (mg)**	**Vitamin B1 (mg)**	**Vitamin B12 (µg)**	**Folate (µg)**	**Vitamin B6 (mg)**	**Vitamin C (mg)**
7.7 ± 4	0.8 ± 0.3	5.7 ± 6.5	284.8 ± 148.1	1.6 ± 0.6	131.1 ± 92.4

± standard deviation.

**Table 4 nutrients-12-02621-t004:** Composition of the microbiota.

Genus	Relative % Abundance (Cutoff > 1)
*Haemophilus*	1.10
*Lachnospiracea_incertae_sedis*	1.26
*Phascolarctobacterium*	1.35
*Veillonella*	1.42
*Lactobacillus*	1.64
*Roseburia*	1.87
*Faecalibacterium*	1.88
*Blautia*	1.95
*Clostridium sensu stricto*	2.05
*Streptococcus*	2.19
*Oscillibacter*	2.62
*Barnesiella*	2.68
*Escherichia/Shigella*	4.31
*Parabacteroides*	4.69
*Alistipes*	9.71
*Prevotella*	12.29
*Bacteroides*	26.40

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
