# Peer review of "Can the FUT 2 Gene Variant Have an Effect on the Body Weight of Patients Undergoing Bariatric Surgery?—Preliminary, Exploratory Study"

_nutrients, 2020, doi:10.3390/nu12092621_

Round 1

Reviewer 1 Report

Introduction is not enough clear and should be rewritten, especially the relations between micorbiota and weight loss effects at large.

Line 72-73: it seems that all patients included had some depressive disorder. If so, it is never stated in text whether they were on antidepressant drugs. This should be clearly reported, given their well known orexygenic effect.

Line 149-150: roughly 30% of patients had symptoms that could somehow impair "regular feeding" and microbiota. No mention of this finding in the discussion. 

Author Response

We thank the reviewers for the very careful evaluation of our manuscript and we are grateful for valuable comments. We have uploaded the most recent version of the manuscript. All changes are highlighted using red color (Reviewer 1) and blue color (Reviewer 2) in Microsoft Word. We were encouraged to address the following points:

Reviewer 1

  1. Introduction is not enough clear and should be rewritten, especially the relations between microbiota and weight loss effects at large.

It was corrected.

We added information about number of bariatric surgeries performed, as follow: The number of patients with severe obesity is gradually increasing. With relation to conservative treatments, bariatric surgery is considered to be a more effective way of treating patients with severe obesity leading to reduction in both weight and coexisting diseases and mortality [1].  According to American Society for Metabolic and Bariatric Surgery the number of metabolic and bariatric procedures performed in 2018 was 10.8% higher than in 2017, which means that only approximately 1% of all patients who qualify for bariatric surgery actually undergo surgery [2].

We also added the following text to clarify the impact of microbial changes after bariatric surgery on weight loss: In systematic review have been shown that bariatric surgeries affect human microbiome. Microbial richness increases significantly within 3 months and maintain after 1 to 2 years post-surgery. Greater alterations in the microbiome occurred after Roux-en-Y gastric bypass, than after purely restrictive bariatric procedures (e.g. sleeve gastrectomy). An increase in Bacteroidetes and Proteobacteria and a decrease in Firmicutes post-bariatric surgery are observed. These changes induce an increase of the Bacteroides/Firmicutes ratio which are positively associated with weight loss post-bariatric surgery.

  1. Line 72-73: it seems that all patients included had some depressive disorder. If so, it is never stated in text whether they were on antidepressant drugs. This should be clearly reported, given their well-known orexigenic effect.

It was added. All participants declared that they did not take antidepressants.

  1. Line 149-150: roughly 30% of patients had symptoms that could somehow impair "regular feeding" and microbiota. No mention of this finding in the discussion. 

It was added. It is also important that although patients were on average 3.2 years after surgery, their BMI and WHR indices still indicated visceral obesity (I degree) and depressive disorders. The diet used by the patients - high-protein, at the same time poor in dietary fiber changes the character of bacterial fermentation and leads to intestinal dysbiosis [48]. Symptoms occurring in almost 30% of our patients (such as recurrent abdominal pain, diarrhea, constipation) may be a way to manifest disorders within the intestinal microflora what was previously described after RYGB [49].That disorders negatively impact health-related quality of life because of its effect on gastrointestinal motility, visceral hypersensitivity, changes in gut permeability, immune activation, gut-brain dysregulation and central nervous system dysfunction [50].

Reviewer 2 Report

This is a peer review of a manuscript entitled, “Can the FUT2 gene variant have an effect on the body weight of patients undergoing bariatric surgery? – preliminary study” by Komorniak and colleagues. The contribution of the gut microbiome to the metabolic improvements observed after bariatric and metabolic surgery has increasingly been identified in the last several years, however, most of these studies are largely observational and correlative without much understanding of mechanism.

In the current study the authors describe a patient population who have previously had gastric bypass (RYGB) and examine their characteristics after being allocated to either the ‘secretor’ phenotype or ‘non-secretor’ phenotype. Overall this is a very interesting study, but the data are clearly preliminary/pilot. The findings are largely correlative and the discussed mechanisms speculative at best, although many of the studies in the field are currently because of the limitations of these techniques. There is more detail that needs to be gathered about the patients included in the study – these details include the numbers of operating surgeons, potential differences in technique, other operations since RYGB that may alter the metabolic effects (i.e. revisional procedures). Many of these details are important and may be difficult to collect, but their absence is important to note. The data overall are IMPORTANT to this nascent field and I hope the authors can clarify the preliminary nature of the findings and add some of the important details that are lacking in the patient population/study design. My comments are below.

Comments

Se/Se = Secretor phenotype (? GG Variant – that should be specifically noted/clarified in the abstract) = propionate content and increased Prevotella, Escherichia, Shigella, Bacteroides

Non-Secretors (GA/AA Variants) = higher abundance of Enterobacteriaceae, Ruminococcaceae, Enterobacteriaceae, Clostridiales

Conclusions by authors = increased [propionate] among GG variant may lead to additional source of energy for the patient… increased risk of WHR cf. other variants (GA/AA)

Introduction

  1. The first sentence indicates that obesity is increasing which in turn is leading to an increasing frequency of bariatric surgery. This is not true, though the number of bariatric operations being performed has slowly continued to rise in the last several years [1]. Bariatric surgery continues to be an underutilized treatment of obesity with ~1% or less of patients eligible actually having surgical intervention. Please correct.
  2. Third paragraph – it would be good to indicate ‘what’ is being secreted in the patients with at least one Se allele.

Methods

  1. You indicate that the study included patients that had depressive disorders. There is no mention, however, of whether this patient population was treated/untreated and what, if any, anti-depressive medications might have be used by the subjects. Given that antidepressant medications can have effects on body weight/diabetes status this is important information. If this is unknown it is critical to mention this in the limitations.
  2. Recruitment = what was the population that was recruited? There is no overall ‘n’ for recruitment… you just list that n=117 were recruited (from what ‘n’ of larger population?)... How was the recruiting done? Where had the patients had surgery? Were more than one surgeon’s patients represented? Were the operations done similarly? These are all important details.
  3. What R packages were used for the PCA? This needs to be indicated for reproducibility.
  4. What statistical tests were used for comparisons in the manuscript? This is not noted. Were the populations normally distributed?
  5. Stats/Power = was this a pilot study or was power calculated a priori? If power was calculated it needs to be explained/noted. Please clarify/cite as able.

Results

  1. Line 175 – you note “there was a significant correlation (p=0.01) between abundance of OTUs…” – was this actually a correlation that was performed? It would be much more powerful to use regression to show the strength of the ‘correlation’ in this case. Please re-write using regression parameters… including a 95% CI, P-value, and point estimate for the variable of interest. These are parameters that are easily calculated with simple linear regression.

Discussion

  1. First paragraph is much too long – please divide up for reader clarity.
  2. What is the limitation of using WHR compared to body mass index or even just raw body weight? Change is presented in the results, but could these be graphically shown in terms of pre- post- surgical body weight in each of the groups (i.e. Secretor vs. Non-Secretor)?
  3. Please add a paragraph focusing on the limitations of the current study.

Tables

  1. Tables – please note the units and what the +/- indicate for each measurement. Please define all abbreviations in the tables.
  2. Table 2 = please list pre- and post-operative BMI in addition to the delta. This is important as significant variation can be missed when looking at change only instead of raw values.
  3. Table 4 = needs units on Y-axis.
  4. Table 5 = needs increase in font size on axes.

References

[1] English WJ, DeMaria EJ, Hutter MM, Kothari SN, Mattar SG, Brethauer SA, et al. American Society for Metabolic and Bariatric Surgery 2018 estimate of metabolic and bariatric procedures performed in the United States. Surg Obes Relat Dis Official J Am Soc Bariatr Surg 2020. https://doi.org/10.1016/j.soard.2019.12.022.

Author Response

We thank the reviewers for the very careful evaluation of our manuscript and we are grateful for valuable comments. We have uploaded the most recent version of the manuscript. All changes are highlighted using red color (Reviewer 1) and blue color (Reviewer 2) in Microsoft Word. We were encouraged to address the following points:

Reviewer 2

Se/Se = Secretor phenotype (? GG Variant – that should be specifically noted/clarified in the abstract) = propionate content and increased Prevotella, Escherichia, Shigella, Bacteroides. Non-Secretors (GA/AA Variants) = higher abundance of Enterobacteriaceae, Ruminococcaceae, Enterobacteriaceae, Clostridiales. Conclusions by authors = increased [propionate] among GG variant may lead to additional source of energy for the patient… increased risk of WHR cf. other variants (GA/AA)

It was corrected.

Introduction

  1. The first sentence indicates that obesity is increasing which in turn is leading to an increasing frequency of bariatric surgery. This is not true, though the number of bariatric operations being performed has slowly continued to rise in the last several years [1]. Bariatric surgery continues to be an underutilized treatment of obesity with ~1% or less of patients eligible actually having surgical intervention. Please correct.

It was corrected. According to American Society for Metabolic and Bariatric Surgery the number of metabolic and bariatric procedures performed in 2018 was 10.8% higher than in 2017, which means that only approximately 1% of all patients who qualify for bariatric surgery actually undergo surgery.

  1. Third paragraph – it would be good to indicate ‘what’ is being secreted in the patients with at least one Se allele.

It was corrected. Individuals with at least one Se allele (Se/Se or Se/se genotypes) have ABO and H system antigens on epithelial cells and are called secretors (variant G428A (se428, W143X, rs601338). In contrast, homozygotes for FUT2 AA (se/se, The FUT2-AAA (G428A, W143X)), are called non-secretors [6–8], which means they are unable to secrete histo-blood group antigens into bodily fluids, or express them on mucosal surfaces [9].

Methods

  1. You indicate that the study included patients that had depressive disorders. There is no mention, however, of whether this patient population was treated/untreated and what, if any, anti-depressive medications might have been used by the subjects. Given that antidepressant medications can have effects on body weight/diabetes status this is important information. If this is unknown it is critical to mention this in the limitations.

It was added. All participants declared that they did not take antidepressants.

  1. Recruitment = what was the population that was recruited? There is no overall ‘n’ for recruitment… you just list that n=117 were recruited (from what ‘n’ of larger population?). How was the recruiting done? Where had the patients had surgery? Were more than one surgeon’s patients represented? Were the operations done similarly? These are all important details.

It was specified. Patients were recruited at the Surgical Obesity Surgery Outpatient Clinic in Szczecin Zdunowo, Poland, in the period from July 2018 to March 2019. Of the outpatient clinic, all patients (n=117) who underwent Sleeve gastrectomy or Roux-en-Y gastric bypass surgery in at least last 6 months were asked to fill in the Beck scale. Out of 117 interviewed individuals, 49 persons complied with the criteria for inclusion (Beck scale ≥ 12 points) in the study, of which 19 women participated in the study.

The answers to the other questions are set out in paragraph 'Limitations', as follow:

This study has several limitations worth noting. Although we stated that the number of patients who complied the inclusion criteria were 49, only 19 people were enrolled in this research. This undoubtedly makes it necessary to repeat the study on a larger group of participants. Second, the patients were recruited at the Surgical Obesity Surgery Outpatient Clinic in Szczecin Zdunowo but due to its outpatient nature, patients who have been operated on in different hospitals may be treated here (in this research 11 patients were operated on by the two surgeons from Zdunowo Hospital and 8 patients by other surgeons). Thirdly, although all patients underwent laparoscopic SG or RYGB bariatric surgery, they were operated on by different surgeons, which creates a risk of potential differences in surgical techniques that could affect the results of this study.

  1. What R packages were used for the PCA? This needs to be indicated for reproducibility.

It was added. Principal component analysis (PCA), non-metric multidimensional scaling (NMDS) and redundancy analysis (RDA) ordinations on Bray-Curtis dissimilarities were performed in R- 3.5.1 [28], package (VEGAN).

  1. What statistical tests were used for comparisons in the manuscript? This is not noted. Were the populations normally distributed?

It was added. Due to small sample size nonparametric analyses were adopted as appropriate. To analyze whether the FUT genotype affect weight loss, a Kruskal-Wallis test was used.

  1. Stats/Power = was this a pilot study or was power calculated a priori? If power was calculated it needs to be explained/noted. Please clarify/cite as able.

A priori sample size calculation was not performed, as no association between FUT genetic variations, microbiome and weight loss have been conducted so far.

Results

  1. Line 175 – you note “there was a significant correlation (p=0.01) between abundance of OTUs…” – was this actually a correlation that was performed? It would be much more powerful to use regression to show the strength of the ‘correlation’ in this case. Please re-write using regression parameters… including a 95% CI, P-value, and point estimate for the variable of interest. These are parameters that are easily calculated with simple linear regression

The p-value reported here is obtained from envfit function (vegan R package), the function fits environmental vectors or factors onto ordination, the current model was performed with 10000 permutations i.e. how many random permutations to test for assigning significance to the fitted vectors. Thus, vectors were fitted onto ordination scores of OTU abundance data, not the OTU abundance data directly. We feel that this approach is much better for this type of data than regression models per OTU. The vectors that were significant (p-value <.05) are shown on Figure 8 as blue arrows. We will clarify by rewriting – ‘there was a significant correlation between abundance of OTU’s ordination with two categorical factors…’.

Discussion

  1. First paragraph is much too long – please divide up for reader clarity.

It was corrected. The discussion was divided into 4 parts.

  1. What is the limitation of using WHR compared to body mass index or even just raw body weight? Change is presented in the results, but could these be graphically shown in terms of pre- post- surgical body weight in each of the groups (i.e. Secretor vs. Non-Secretor)?

As was mentioned in the article WHR is a better (more precise) indicator than BMI and body weight. This is because WHR takes into account the body composition (the proportion between fat and fat-free body weight). In the context of health risks associated with excess body fat (especially metabolically active visceral fat) WHR is a more precise indicator. Studies have shown that WHR is a better predictive factor for visceral fat, cardiometabolic diseases and mortality than BMI.

We added a Figure 2 Body weight in case of FUT2 variants

  1. Please add a paragraph focusing on the limitations of the current study.

It was added.

Limitations

This study has several limitations worth noting. Although we stated that the number of patients who complied the inclusion criteria were 49, only 19 people were enrolled in this research. This undoubtedly makes it necessary to repeat the study on a larger group of participants. Second, the patients were recruited at the Surgical Obesity Surgery Outpatient Clinic in Szczecin Zdunowo but due to its outpatient nature, patients who have been operated on in different hospitals may be treated here (in this research 11 patients were operated on by the two surgeons from Zdunowo Hospital and 8 patients by other surgeons). Thirdly, although all patients underwent laparoscopic SG or RYGB bariatric surgery, they were operated on by different surgeons, which creates a risk of potential differences in surgical techniques that could affect the results of this study.

Tables

  1. Tables – please note the units and what the +/- indicate for each measurement. Please define all abbreviations in the tables.

It was corrected.

  1. Table 2 = please list pre- and post-operative BMI in addition to the delta. This is important as significant variation can be missed when looking at change only instead of raw values.

This information was added to table 1. BMI before surgery 43,2 (±5,9) and after 30,6 (±5,3).

  1. Table 4 = needs units on Y-axis.

It was added. After the amendments, it is figure 5 and y-axis is relative percent abundance.

  1. Table 5 = needs increase in font size on axes.

It was corrected. We've increased the size of the scheme (after the amendments, it is figure 6).

Round 2

Reviewer 2 Report

I appreciate the authors corrections and clarifications from the first version. This is an interesting study and merits publication. My only comment is that in the methods it must explicitly be stated that this is an EXPLORATORY study with no a prior power calculation. This has significant implications for the findings and notes the necessary confirmation of these findings in properly powered future studies.

Author Response

We thank the reviewer for the very careful evaluation of our manuscript and we are grateful for valuable comments. I have uploaded the most recent version of the manuscript. All changes are highlighted using yellow color:

 Rev.2 I appreciate the authors corrections and clarifications from the first version. This is an interesting study and merits publication. My only comment is that in the methods it must explicitly be stated that this is an EXPLORATORY study with no a prior power calculation. This has significant implications for the findings and notes the necessary confirmation of these findings in properly powered future studies.

I added world exploratory for tittle and now is:

Can the FUT 2 gene variant have an effect on the body weight of patients undergoing bariatric surgery? – preliminary, exploratory study.

In methodology,  line 91 It  was added:  This was an exploratory study without a prior power calculation. 

In limitation, line 295 it was added  This was an exploratory study without a prior power calculation. Therefore  is the necessary to confirm of these findings in next, properly powered future studies.
